# Re-Irradiation by Stereotactic Radiotherapy of Brain Metastases in the Case of Local Recurrence

**DOI:** 10.3390/cancers15030996

**Published:** 2023-02-03

**Authors:** Ruben Touati, Vincent Bourbonne, Gurvan Dissaux, Gaëlle Goasduff, Olivier Pradier, Charles Peltier, Romuald Seizeur, Ulrike Schick, François Lucia

**Affiliations:** 1Radiation Oncology Department, University Hospital, 29200 Brest, France; 2LaTIM INSERM UMR 1101, University of Western Brittany, 29200 Brest, France; 3Neurosurgery Department, University Hospital, 29200 Brest, France

**Keywords:** stereotactic radiation therapy, brain metastases, re-irradiation, local control, radionecrosis

## Abstract

**Simple Summary:**

This is the first study to evaluate the outcomes of repeated salvage stereotactic radiotherapy (SRT) for the local recurrence of brain metastases after initial stereotactic radiotherapy at our center using the Hypofractionated Treatment Effects in the Clinic (HyTEC) reporting standards of the WGSBRT and the European Society for Radiotherapy and Oncology guidelines. The performance of surgery was predictive of a significantly better local control and survival. The volume of a normal brain receiving 5 Gy during re-irradiation, a dose delivered to the PTV in the first irradiation, and concomitant systemic therapy were associated with an increased risk of radionecrosis.

**Abstract:**

Purpose: To evaluate the efficacy and safety of a second course of stereotactic radiotherapy (SRT2) treatment for a local recurrence of brain metastases previously treated with SRT (SRT1), using the Hypofractionated Treatment Effects in the Clinic (HyTEC) reporting standards and the European Society for Radiotherapy and Oncology guidelines. Methods: From December 2014 to May 2021, 32 patients with 34 brain metastases received salvage SRT2 after failed SRT1. A total dose of 21 to 27 Gy in 3 fractions or 30 Gy in 5 fractions was prescribed to the periphery of the PTV (99% of the prescribed dose covering 99% of the PTV). After SRT2, multiparametric MRI, sometimes combined with 18F-DOPA PET-CT, was performed every 3 months to determine local control (LC) and radionecrosis (RN). Results: After a median follow-up of 12 months (range: 1–37 months), the crude LC and RN rates were 68% and 12%, respectively, and the median overall survival was 25 months. In a multivariate analysis, the performance of surgery was predictive of a significantly better LC (*p* = 0.002) and survival benefit (*p* = 0.04). The volume of a normal brain receiving 5 Gy during SRT2 (*p* = 0.04), a dose delivered to the PTV in SRT1 (*p* = 0.003), and concomitant systemic therapy (*p* = 0.04) were associated with an increased risk of RN. Conclusion: SRT2 is an effective approach for the local recurrence of BM after initial SRT treatment and is a potential salvage therapy option for well-selected people with a good performance status. Surgery was associated with a higher LC.

## 1. Introduction

Throughout the course of their illness, brain metastases (BMs) will appear in about 30% of cancer patients [1]. Advances in medical oncology have been associated with longer extra-cranial progression-free survival (PFS) and overall survival (OS), resulting in higher accrual rates of BMs. In the absence of treatment, the prognosis is usually poor, with OS lasting only a few weeks or months, with often an impaired quality of life [2]. Whole brain radiotherapy (WBRT) has long been the standard radiotherapy modality for the treatment of BMs. Alongside surgery and systemic treatments, stereotactic radiosurgery (SRS) or fractionated stereotactic radiotherapy (SRT) now plays an important role in improving local control (LC) while decreasing treatment-related toxicity. Local failure (LF) following SRT or SRS is still relatively rare, with reported rates of LC at one year being greater than 90% [3,4]. The management of post-SRT/-SRS LF remains troublesome [5,6,7,8,9] in an era of improved OS thanks to new treatment modalities such as targeted therapy and immune check point inhibitors [10,11,12,13]. Surgery as a stand-alone treatment has a significant rate of LF and may be challenging if lesions are deeply located or in functional areas [14]. WBRT carries a substantial risk of neurocognitive side effects and, when used alone, does not result in lasting LC [15]. With a median LC of 2 ± 4 months and an OS of 3 ± 7 months, chemotherapy has poor outcomes for these individuals as well [16,17].

However, numerous studies have demonstrated that subsequent SRS/SRT treatments can be delivered with excellent efficacy and tolerance for the management of new distant BM following a first course of SRS/SRT in order to postpone WBRT [18,19]. Stereotactic re-irradiation (SRS2/SRT2) appears as an effective management strategy for LF after SRS/SRT too [20]. SRS2/SRT2 have been used more frequently in patients with LF over the past decade as a result of these encouraging findings and the paucity of suitable alternatives. The effectiveness and tolerability of this approach are debatable; however, this is primarily due to the lack of standardization in SRS and SRT protocols prior to the publication of the International Commission on Radiation Units and Measurements (ICRU) Report 91, which makes it challenging to compare the outcomes of studies conducted in various centers [21].

In October 2020, based on pooled dosimetric and clinical data from the published literature in English, the American Association of Physicists in Medicine Working Group on Stereotactic Body Radiotherapy (WGSBRT) proposed a predictive model for the probability of tumor control (TCP) after SRS or SRT [22]. The accurate reporting of margin prescription dose, fractionation, and tumor size were recommended to assess their correlation with corresponding actuarial local control and/or overall survival at 1 and/or 2 years. It was also necessary to have at least 10 patients in each stratified group. Only 56 manuscripts met these eligibility criteria, despite the fact that approximately 3000 potentially relevant manuscripts were reviewed. This demonstrates the discrepancy in the reporting of relevant information in the literature and the need for more systematic reporting of data in this particular clinical setting.

The purpose of our study was to analyze the safety and efficacy outcomes of repeated salvage SRT for the local recurrence of brain metastases after initial SRT at our center using the Hypofractionated Treatment Effects in the Clinic (HyTEC) reporting standards of the WGSBRT [22,23] and the European Society for Radiotherapy and Oncology guidelines [24].

## 2. Materials and Methods

### 2.1. Patients’ Selection

Patients treated at least twice with SRT on the same brain metastasis between December 2014 and May 2021 at our institution were retrospectively considered for this study. The decision to perform a second course of SRT on a previously treated lesion was made by an experienced multidisciplinary board. As there is no gold standard for the differential diagnosis between radionecrosis and tumor progression, the diagnosis of tumor progression was assessed using contrast-enhanced T1-I, T2-weighted, and perfusion MRIs. If a multiparametric MRI was inconclusive, a 3,4-dihydroxy-6-(18)F-fluoro-L-phenylalanine (^18^F-DOPA) positron emission tomography (PET)-CT scan was performed. Thus, the diagnosis of a local recurrence after SRT1 was made on multiparametric MRI features including an increased contrast uptake at the treated site, elevated relative cerebral blood volume with perfusion, elevated peak choline with spectroscopy, hypermetabolism with ^18^F-DOPA positron emission tomography, and/or biopsy/resection demonstrating residual tumor cells.

A signed authorization for the use of their clinical data for scientific purposes and informed consent for anonymous publication of the data was obtained for each patient. An institutional review board approved this study.

### 2.2. Baseline Evaluation and Treatment

Patient characteristics including age; gender; histological findings; extracranial disease status; the Karnofsky performance scale (KPS) score; previous WBRT; previous surgery; and the use of systemic treatments including chemotherapy, hormone therapy, immunotherapy or targeted therapies, the control of primary disease, recursive partitioning analysis (RPA classification), and the disease-specific grade prognostic assessment (DS-GPA) were collected from their medical records. Outcomes including local control, toxicity, and radionecrosis were also determined via the electronic medical records. A slice thickness of 1.5 mm was used for the acquisition of the planning CT (Siemens©, Somatom, Forchheim, Germany). Patient immobilization was performed with a frameless thermoplastic mask (BrainLAB©, Feldkirchen, Germany). In order to define the gross tumor volume (GTV), the macroscopic contrast-enhancing lesion on a T1 MRI and the organs at risk (brain, normal brain minus GTV/PTV, eyes, lens, optic chiasm, optic nerves, brainstem, inner ear, and spinal cord), a co-registration of planning CT and T1-MRI sequences, usually a three-dimensional spoiled gradient series with a slice thickness of 1 mm performed within the last 2 weeks before the start of the SRT, was performed.

A total dose of 21 to 27 Gy in 3 fractions or 30 Gy in 5 fractions was prescribed at the periphery of the PTV (99% of the prescribed dose covering 99% of the PTV). The dose was decided according to various clinical and radiological parameters, including the size or volume of the BM, the presence of subacute or acute neurological symptoms, the proximity of the OARs, and the critical anatomical position at each course of SRT.

Two of the treating physicians (RT and FL) independently reviewed each record, including all neuroradiological reports and corresponding images and plans of these patients.

We reported all target and brain dose metrics according to the Hypofractionated Treatment Effects in the Clinic (HyTEC) reporting standards and the European Society for Radiotherapy and Oncology guidelines. Dosimetric parameters were extracted from our treatment planning system (Philips© Pinnacle version 16.2). In accordance with these reporting standards, we calculated the percentage of the prescription isodose line by dividing the prescribed dose by the maximum dose. We also reported the biologically effective dose (BED) using the formula
BED = D × (1 + [d / (α/β)])(1)
where the variables are as follows: d = dose per fraction, in Gy; D = total dose (number of fractions × dose per fraction), in Gy; and α/β ratio = the property of irradiated tissue. 

We reported physical and biological cumulative doses after a rigid registration of treatment planning.

### 2.3. Outcome Evaluation

Patient follow-up consisted of serial MRIs (same as for treatment planning), first 6 weeks after SRS, then at 3 months, and then every 3 months. Intermediate imaging could also be performed if medically indicated. We used the same criteria for assessing the response to SRT2 as for SRT1.

### 2.4. Statistical Methods

To describe the general behavior of the data, we used standard descriptive statistics. LC was defined from the first day of SRT2 to the time of local relapse. Overall survival (OS) was calculated from the start of SRT2 to the time of death or last follow-up date. The log-rank test or univariate Cox regression was used, respectively, for categorical and numerical data to assess the prognostic role of individual variables. To determine the threshold values for significant parameters, the receiver operating characteristic (ROC) curve was used with the Youden index. The multivariate Cox model was used as a method to estimate the independent association of a variable set with OS and LC. The prognostic factors analyzed were clinical, dosimetric parameters, and the administration of systemic therapies (including targeted therapy, immunotherapy, and chemotherapy) after local treatments of BMs. We used a 1-month delay before or after SRT to define a systemic treatment as concomitant, considering the date of the start of SRT if the systemic treatment was administered before, and the date of the end of SRT was taken into account if patients started systemic therapy after radiation [25,26]. Regarding the tumor volume, we considered the largest GTV and PTV in the case of several irradiated BMs. We also calculated the sum of the GTVs (and PTVs) and evaluated the correlation with the outcome. All statistical analyses were performed using the R++ platform for statistical programming, version 1.5.03.

## 3. Results

### 3.1. Patient and Tumor Characteristics

Between December 2014 and May 2021, 733 brain SRTs were realized in our institution. Among them, we identified 32 patients with 34 BMs initially treated with SRT (SRT1) that developed local recurrence and underwent a repeated course of SRT (SRT2) on the same site.

The patients and tumor characteristics are summarized in Table 1. There were 17 men and 15 women for a total of 34 lesions. Patients received SRT2 for a single metastasis (*n* = 31) or for three metastases (*n* = 1). The median time interval between SRT treatments was 12 months (range: 3–65 months). Nineteen patients had extracranial disease at the time of re-irradiation. The median GTV volume was 3.8 cc (range: 0.14–67.5 cc), and the median PTV volume was 8.3 cc (range: 0.7–97.8 cc). The median delivered dose (BED with an α/β ratio of 10) to 100% of the GTV during the second course of SRT was 40.89 Gy (range: 24.87–50.36 Gy). The cumulative median delivered dose (BED) to 100% of the GTV was 81.78 Gy (range: 76.59–91.25 Gy).

The medians of V5Gy, V10Gy, V15Gy, V20Gy, and V25Gy were 10.2, 4.2, 2.2, 1.2, and 0.5 cm^3^, respectively. The dosimetric parameters for SRT1, SRT2, and cumulative SRT are summarized in Table 2, Appendix A. Seventeen patients were analyzable at one year (thirteen deaths and two lost to follow-up), and five patients were analyzable at two years (sixteen deaths and eleven lost to follow-up).

### 3.2. Local Control

With a median follow-up study of 12 months (range: 1–37 months), the crude local control was 68%, with the 1- and 2-year local control rates being at 60% and 25%, respectively (Figure 1). In the univariate analysis, NSCLC histology (*p* = 0.005) and a lack of surgery (*p* < 0.0001) (Figure 1) were associated with worse LC; however, only the lack of surgery before SRT2 was predictive of local failure in the multivariate analysis (*p* = 0.002; HR 12.8, 95% CI: 1.5–110.0) (Table 3). The 1-year local control rates varied depending on the histology with the rates of 80%, 63%, and 38% for melanoma (*n =* 5), breast carcinoma (*n =* 6), and NSCLC (*n =* 16) metastases, respectively. No other factors, including neither the volume of metastases at the time of SRT2 nor the interval between the first and the second courses of SRT, were significantly predictive of local control.

### 3.3. Overall Survival

The median survival was 25 months, and the 1- and 2-year survival rates were 60% and 50% (Figure 2), respectively. Twenty-five (58%) patients succumbed to their extracranial disease and eleven (26%) to their brain disease. Based on the univariate analysis, KPS (*p* = 0.05), the performance of surgery before SRT2 (*p* = 0.006) (Figure 2), and a lower V25 in the normal brain, were associated with a better OS. After the multivariate analysis, only the performance of surgery was predictive of a significant survival benefit (*p* = 0.04; HR 6.8, 95% CI: 1.1–61.02) (Appendix A).

### 3.4. Toxicity

Four cases of radionecrosis out of thirty-four treated lesions (12%) were found with multiparametric MRI and PET imaging, with a median time to onset of six months (range: 4–10 months). Of these four patients, two required treatment with high-dose dexamethasone because of neurological complications. The actuarial risk of radionecrosis after repeated SRT was 25% at 1 and 2 years for both. In the univariate analysis, the factors significantly associated with the development of brain radionecrosis were the volume of a normal brain receiving 5 Gy at SRT2, the delivered doses to PTV during SRT1, and SRT2, a concomitant systemic treatment. In the multivariate analysis, the volume of a normal brain receiving 5 Gy during repeated SRT (*p* = 0.04), the delivered doses to PTV during SRT1 (*p* = 0.003), and a concomitant systemic treatment (*p* = 0.04) were predictive of radionecrosis (Table 4).

## 4. Discussion

The management of the local recurrent BM after a first course of SRT remains challenging. Data on the optimum salvage treatment method for in-site recurrent BMs following SRS/SRT are lacking. Surgical excision, systemic therapy, or re-irradiation with WBRT or SRS2/SRT2 are all alternatives for treatment. A variety of criteria influence the decision, including the patients’ age and functional level, intracranial tumor burden and localization, control of extracranial disease, prior treatments, primary cancer type, and prospect of targeted therapy [6]. When it is possible to identify tumor recurrence from radionecrosis, surgery is the preferred option. Post-operative LC rates greatly vary depending on the studies, ranging from 62% to 93% at one year [7,9,27] and a median survival of 8.7 months [7]. Even with a neurosurgical technique, re-irradiation is frequently required to obtain improved LC [14]. Moreover, surgery is rarely performed, with only 1–11% of patients requiring salvage treatment for recurrent BMs being operated on, due to its invasiveness and morbi-mortality [8,27,28]. On the other hand, because of limited blood–brain barrier penetrability, systemic therapy remains poorly effective in the management of most brain metastases [29]. Finally, because normal brain tissue was thought to be at risk of irreparable tissue damage, the re-irradiation of central nervous system malignancies was long deemed unwise. The use of WBRT raises the risk of eventual durable cognitive impairment and reduces patient quality of life, especially in long-term survivors who are oligometastatic or have solely intracerebral progression [15,30,31]. Furthermore, because local recurrence following high-dose SRS/SRT therapy is frequently regarded as a radioresistant lesion, lower doses of WBRT than those of SRS/SRT are unlikely to provide long-term disease control. As a result, radiation oncologists are still hesitant to re-irradiate the brain with conventional radiation treatment approaches.

Localized irradiation appears to be an attractive prospect since it may have a lower toxicity profile, and the re-irradiation of a local target by SRS/SRT is of interest with a higher conformation to the target volume. This technique has already demonstrated its efficacy and tolerability in other diseases, such as vestibular schwannomas and meningiomas [32,33]. The prescribed doses, however, are lower than those used for BM re-irradiation.

To our knowledge, 13 retrospective studies investigated the use of a second course of SRS/SRT treatment for recurrent BMs, with a 1-year LC rate of 72.5% and a 1-year OS rate of 54% [20].

Our study reported on a series of 32 patients who underwent a second course of SRT for 34 recurrent BMs at the site of primary SRT. At one-year, the local control and overall survival rates were 60%, which was consistent with previous studies. However, these results were inferior to those obtained after primary SRT. One hypothesis is that the majority of treated lesions can be considered radioresistant because they recurred after a first course of SRT. Surgery before SRT2 could reduce the tumor burden, which would explain the significant benefit, both in terms of the LC and OS of surgery in our study [34]. These results also underline the importance of a multidisciplinary discussion, including experienced neurosurgeons, radiation oncologists, and radiologists, for the management of local recurrences.

Otherwise, we found that the local control rates at 1 year varied by histology, with rates of 80%, 63%, and 38% for melanoma (*n =* 5), breast carcinoma (*n =* 6), and NSCLC (*n =* 16) metastases, respectively. These findings may appear surprising, but they could be related to the systemic treatment. Indeed, in our series, three out of five melanoma metastases concerned the same patient, who responded perfectly to immunotherapy. 

Radionecrosis is the major side effect after stereotactic radiotherapy. Indeed, its rate varies according to the studies in the literature between 4% and 24% [35,36,37,38]. Radiation dose, fractionation, and subsequent chemotherapy administration are known to be the main risk factors [39]. In our study, the rates of toxicity we observed seem to be more acceptable in comparison with the possible neurocognitive deterioration and decreased quality of life seen particularly in long-term WBRT survivors [4,31]. Our data suggested that the volume of a normal brain receiving 5 Gy during repeated SRT, the delivered doses to PTV during SRT1, and a concomitant systemic treatment may be predictive of radionecrosis. This must be explored, and people should consider the possibility of reducing PTV margins, subject to strict adherence to the constraint used. More research is needed to thoroughly evaluate this option.

Although our study had a limited number of patients, data regarding the specific issue of repeated fractionated SRT on a BM after initial SRT failure are limited to retrospective studies, with a similar sample size between 30 and 47 patients [20].

Moreover, the lack of standardization in data collection was significant in previous studies. To our knowledge, our study was the first to use the HYTEC reporting standards of the WGSBRT and the European Society for Radiotherapy and Oncology guidelines [24], allowing a consistent comparison with other future trials.

Given the relatively low frequency of local failure after SRT, this retrospective report was limited by its small sample size and patient selection bias, and this must be taken into account when evaluating safety and viability. The small number of patients treated exclusively with volumetric modulated arc therapy and the single-institutional retrospective research design also restrict the interpretation of our findings. Multiple biases in data collection may have resulted in an underestimation of radionecrosis risk, and it is probable that some patients with radionecrosis were missed due to a lack of follow-up. Moreover, the use of several doses and fractionation regimens limited our ability to identify specific variables that may contribute to local control and the development of radionecrosis, but reporting the doses using the BED reduced this bias. Finally, one limitation of the findings on the influence of surgery on LC and OS was that we did not have data on the quality of surgical resection, whether complete or partial. However, despite these limitations, we believe that this study adds to the sparse data on repeated SRT for in-field recurrence and provides new insights into an increasingly common clinical setting. To assess the role of SRS2/SRT2 for patients with recurrent brain metastases in carefully chosen patients, additional prospective studies are still required.

## 5. Conclusions

A second course of SRT (SRT2) was a successful method for the local recurrence of BMs following the first SRT treatment, with comparable results in terms of overall survival and LC and to surgical series, and is a viable salvage therapy option for well-selected individuals with a good performance status. If feasible and if the patient’s general condition allows it, a surgery seems attractive before considering performing SRT2. Regardless, it appears essential that each case be discussed in a multidisciplinary committee comprised of an expert neurosurgeon and radiologist for the management of these local recurrences. More research is needed to improve the care of recurrent brain metastases following the first SRT, and prospective trials with a larger sample should be conducted.

## Figures and Tables

**Figure 1 cancers-15-00996-f001:**
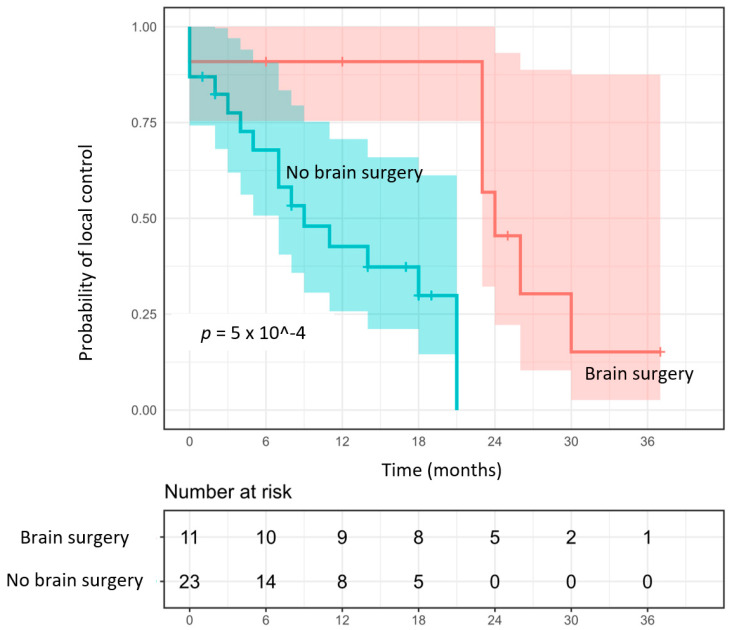
Probability of local control depending on whether or not a surgery was performed before SRT2.

**Figure 2 cancers-15-00996-f002:**
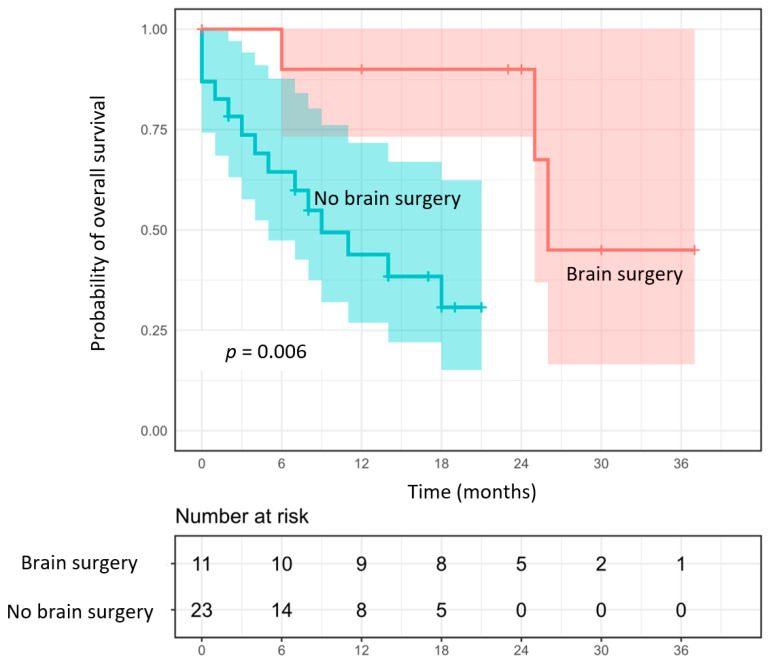
Probability of overall survival depending on whether or not a surgery was performed before SRT2.

**Table 1 cancers-15-00996-t001:** Patients’ characteristics.

Characteristics	Group	
	*n =* 32	%
Gender Male Female	17 15	53 47
Age Median (Range)	63 (42–79)	
Prior WBRT Yes No	5 27	16 84
Primary Breast o HER-2 + Lung o Adenocarcinoma ➢ EGFR + ➢ ALK + ➢ Without Mutation o Squamous o Small-Cell Melanoma Prostate GI Unknown	6 2 18 15 1 1 13 1 2 5 1 1 1	19 6 56 47 3 3 41 3 6 16 3 3 3
Number of BMs Single Multiple (3)	31 1	97 3
Extra Cranial Metastases Yes No	19 13	59 41
DS-GPA Score Median (Range)	2 (1–4)	
KPS Median (Range)	90 (60–100)	
RPA Class I II III	14 16 2	44 50 6
Neurologic Symptoms Yes No	10 22	31 69
Resection Before SRT1 Yes No	6/34 27/34	18 82
Resection Before SRT2 Yes No	11/34 23/34	32 68
Other SRT Yes No	13 19	41 59
Concomitant Systemic Treatment		
Yes	9/34	26
Chemotherapy	4	11
Hormonotherapy	1	3
Immunotherapy	2	6
Anti-HER-2	2	6
Anti-EGFR	0	0
Anti-ALK	0	0
No	25/34	74
Interval Between Treatments (Months) Median (Range)	12	3–65

Abbreviations: WBRT *=* whole brain radiation therapy, HER-2 = Human epidermal growth factor receptor 2**,** EGFR = epidermal growth factor receptor, ALK = Anaplastic lymphoma kinase, GI *=* gastro-intestinal, BMs *=* brain metastases, DS-GPA *=* diagnostic-specific graded prognostic assessment, KPS *=* Karnofsky performance status, RPA *=* recursive partitioning analysis, SRT1 *=* first course of stereotactic radiation therapy, SRT2 *=* second course of stereotactic radiation therapy.

**Table 2 cancers-15-00996-t002:** Dosimetric parameters of second course of SRT.

Dosimetric Parameters	34 Metastases (*n* = 34) Median (Range) %
GTV volume	3.8 (0.14–67.51)
PTV volume median	8.28 (0.68–97.79)
D_min_ GTV	26.09 Gy (21.08–36.14)
D_max_ GTV	32.88 (22.22–41.59)
D_min_ PTV	21.21 (15.03–36.02)
D_max_ PTV	32.88 (22.22–41.59)
Prescription isodose line	81.11% (64.14–97.31)
Target coverage GTV	100 (51.82–100)
Target coverage PTV	98.93 (51.82–99.45)
Delivered dose GTV	23.1 (15.55–39.80)
Delivered dose GTV BED	40.89 (24.87–50.36)
Delivered dose PTV	23.01 (15.55–37.73)
Delivered dose PTV BED	40.71 (24.87–48.00)
V_5_ brain	10.21 (2.35–82.92)
V_10_ brain	4.22 (0.97–54.1)
V_15_ brain	2.15 (0.39–31.70)
V_20_ brain	1.23 (0.00–16.37)
V_25_ brain	0.48 (0.00–8.32)

Abbreviations: GTV *=* gross tumor volume, PTV *=* planning target volume, BED *=* biologically effective dose, V_xGy (brain)_
*=* volume of brain receiving dose larger than or equal to xGy

**Table 3 cancers-15-00996-t003:** Univariate and multivariate analyses for local control (LC).

Variables	Univariate Analysis			Multivariate Analysis		
	HR	95% CI	*p*	HR	95% CI	*p*
Histology (lung vs. others)	3.57	1.43–8.89	0.005	1.66	0.59–4.72	0.34
Age	1.04	0.99–1.09	0.09	1.03	0.98–1.09	0.23
Gender (female vs. male)	0.48	0.19–1.22	0.11	-	-	-
DS-GPA	0.88	0.55–1.42	0.61	-	-	-
RPA	1.67	0.75–3.74	0.21	-	-	-
Number of BMs	1.01	0.78–1.33	0.90	-	-	-
ECM (yes vs. no)	0.86	0.48–1.55	0.61	-	-	-
KPS	0.07	0.01–4.67	0.22	-	-	-
Systemic treatment concomitant	0.83	0.36–1.90	0.66	-	-	-
Chemotherapy (yes vs. no)	0.89	0.31–1.45	0.75	-	-	-
Immunotherapy (yes vs. no)	2.34	0.81–4.11	0.15	-	-	-
Targeted Therapy (yes vs. no)	1.14	0.72–1.61	0.41	-	-	-
GTV volume SRT1	0.98	0.89–1.08	0.72	-	-	-
PTV volume SRT1	0.98	0.92–1.04	0.48	-	-	-
GTV volume SRT2	1.00	0.98–1.02	0.94	-	-	-
PTV volume SRT2	1.00	0.99–1.01	0.91	-	-	-
Interval treatment time	1.03	0.98–1.08	0.32	-	-	-
Brain surgery SRT2 (yes vs. no)	19.47	2.36–160.37	<0.0001	12.76	1.48–110.01	0.002
Brain surgery SRT1 (yes vs. no)	1.02	0.23–4.52	0.98	-	-	-
Delivered dose GTV SRT1	0.47	0.17–1.32	0.23	-	-	-
Delivered dose GTV SRT1 BED	0.74	0.49–1.12	0.23	-	-	-
Delivered dose PTV SRT1	0.86	0.49–1.51	0.61	-	-	-
Delivered dose PTV SRT1 BED	0.90	0.67–1.21	0.50	-	-	-
Delivered dose GTV SRT2	1.00	0.93–1.07	0.94	-	-	-
Delivered dose GTV SRT2 BED	1.01	0.94–1.09	0.69	-	-	-
Delivered dose PTV SRT2	1.01	0.93–1.09	0.88	-	-	-
Delivered dose PTV SRT2 BED	1.03	0.95–1.11	0.45	-	-	-
Delivered dose GTV total	0.96	0.87–1.06	0.42	-	-	-
Delivered dose GTV total BED	0.99	0.89–1.09	0.82	-	-	-
Delivered dose PTV total	0.97	0.89–1.07	0.59	-	-	-
Delivered dose PTV total BED	1.01	0.92–1.11	0.83	-	-	-
Dmax SRT1	0.99	0.88–1.10	0.82	-	-	-
Dmax SRT2	1.01	0.91–1.11	0.88	-	-	-
Dmax total	0.99	0.94–1.05	0.83	-	-	-

Abbreviations: HR *=* hazard ratio, CI *=* confidence interval, ECM *=* extra cranial metastases.

**Table 4 cancers-15-00996-t004:** Univariate and multivariate analyses for radionecrosis (RN).

Variables	Univariate Analysis			Multivariate Analysis		
	HR	95% CI	*p*	HR	95% CI	*p*
Histology (lung vs. others)	1.45	0.20–10.58	0.71	-	-	-
Age	0.94	0.85–1.04	0.22	-	-	-
Gender (female vs. male)	1.09	0.15–7.76	0.93	-	-	-
DS-GPA	1.50	0.47–4.82	0.48	-	-	-
RPA	0.25	0.03–2.23	0.16	-	-	-
Number of BMs	1.37	0.78–2.40	0.30	-	-	-
ECM (yes vs. no)	1.66	0.33–8.21	0.51	-	-	-
KPS	4.09	0.004–37.52	0.76	-	-	-
Concomitant systemic treatment	1.36 × 10^9^	0–inf	0.028	2.22 × 10^9^	0–inf	0.04
Chemotherapy (yes vs. no)	2.89	0.67–8.32	0.09	-	-	-
Immunotherapy (yes vs. no)	2.14	0.43–7.54	0.21	-	-	-
Targeted therapy (yes vs. no)	1.12	0.37–7.71	0.53	-	-	-
Prior WBRT	2.00	0.16–24.33	0.59	-	-	-
GTV volume SRT1	0.88	0.54–1.43	0.53	-	-	-
PTV volume SRT1	0.94	0.73–1.21	0.56	-	-	-
GTV volume SRT2	0.89	0.65–1.22	0.27	-	-	-
PTV volume SRT2	0.93	0.78–1.12	0.27	-	-	-
Interval treatment time	1.00	0.88–1.13	0.98	-	-	-
Brain surgery SRT2 (no vs. yes)	1.19	0.17–8.50	0.86	-	-	-
Delivered dose GTV SRT1	3381.54	0–inf	0.78	-	-	-
Delivered dose GTV SRT1 BED	26.79	0–inf	0.78	-	-	-
Delivered dose PTV SRT1	7.12 × 10^15^	3.56 × 10^−5^–1.42 × 10^36^	0.0002	-	-	-
Delivered dose PTV SRT1 BED	8.98 × 10^8^	0.003–2.61 × 10^20^	0.0002	28,249.20	0.008–1.01 × 10^11^	0.003
Delivered dose GTV SRT2	1.10	0.93–1.31	0.30	-	-	-
Delivered dose GTV SRT2 BED	1.30	0.95–1.79	0.06	-	-	-
Delivered dose PTV SRT2	1.13	0.94–1.36	0.25	-	-	-
Delivered dose PTV SRT2 BED	1.36	0.93–2.00	0.04	1.32	0.49–3.51	0.47
Delivered dose GTV total	1.12	0.90–1.39	0.37	-	-	-
Delivered dose GTV total BED	1.37	0.88–2.13	0.11	-	-	-
Delivered dose PTV total	1.14	0.90–1.44	0.31	-	-	-
Delivered dose PTV total BED	1.44	0.82–2.55	0.09	-	-	-
Dmax SRT1	1.77	0.22–14.30	0.29	-	-	-
Dmax SRT2	0.90	0.71–1.14	0.41	-	-	-
Dmax total	1.10	0.83–1.45	0.50	-	-	-
V_5_ brain SRT1	2.84	0.53–6.35	0.35	-	-	-
V_5_ brain SRT2	3.81	3.58–11.14	0.03	1.74	1.01–4.23	0.04
V_10_ brain SRT1	2.63	0.18–5.25	0.38	-	-	-
V_10_ brain SRT2	2.79	0.46–9.36	0.24	-	-	-
V_15_ brain SRT1	2.52	0.04–8.20	0.56	-	-	-
V_15_ brain SRT2	2.68	0.27–9.75	0.28	-	-	-
V_20_ brain SRT1	2.49	0.006–38.91	0.73	-	-	-
V_20_ brain SRT2	2.61	0.16–8.30	0.34	-	-	-
V_25_ brain SRT1	3.84	0.003–5463.85	0.73	-	-	-
V_25_ brain SRT2	2.60	0.10–6.54	0.46	-	-	-

## Data Availability

Research data are stored in an institutional repository and will be shared upon request to the corresponding author.

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
