# Peer review of "Re-Irradiation by Stereotactic Radiotherapy of Brain Metastases in the Case of Local Recurrence"

_cancers, 2023, doi:10.3390/cancers15030996_

Round 1

Reviewer 1 Report

The study retrospectively evaluates the efficacy and safety of a second course of stereotactic radiotherapy for local recurrence of brain metastases previously treated with SRT, using the hypofractionated Treatment Effects in the Clinic (HyTEC) reporting standards of the WGSBRT and the European Society for Radiotherapy and Oncology guidelines.

The management of recurrent brain metastases after first radiation therapy (RT) is still challenging, and only few retrospective studies investigates the role of a second course of RT. 

It is an interesting study, despite the small sample size and the retrospective analysis. 

Just few minor observations

“Results: Table 1

  • There is an error in the percentage of “prior WBRT”
  • There is an error in the total number of “RPA class”

“Results: Local control

  • In the study, authors evaluated also the concomitant systemic treatment, but it has not been clarified the specific type of treatment: targeted therapy? Immunotherapy? Classical chemotherapy? Actually there are different molecules, usually used in clinical practice for NSCLC, breast cancers and melanomas, that can be active on brain metastases, so it could be useful to specify this point in the text.

“Results: toxicity”

  • Lines 223-224 “The actuarial risk of radionecrosis after repeated SRT was 25 % at 1 and 2 years, respectively”: The percentage is the same at 1year and at 2 years? It is not clear.
  • Line 229 “(…) systemic treatment concomitant (p = 0.04) were predictive of radionecrosis”: authors could clarify if there is a correlation between a specific type of systemic treatment (immunotherapy, targeted therapy, chemotherapy) and the incidence of radiation necrosis and/or symptomatic radiation necrosis.
  • Is there any correlation between radiation necrosis and prior WBRT?

Author Response

“Results: Table 1

  • There is an error in the percentage of “prior WBRT”
  • There is an error in the total number of “RPA class”

We have corrected these 2 mistakes.

“Results: Local control

  • In the study, authors evaluated also the concomitant systemic treatment, but it has not been clarified the specific type of treatment: targeted therapy? Immunotherapy? Classical chemotherapy? Actually there are different molecules, usually used in clinical practice for NSCLC, breast cancers and melanomas, that can be active on brain metastases, so it could be useful to specify this point in the text.

We had already specified in the text that systemic treatment included chemotherapy, immunotherapy, targeted therapy and hormone therapy

« Patient characteristics including age, gender, histological findings, extracranial disease status, Karnofsky Performance Scale (KPS) score, previous WBRT, previous surgery, use of systemic treatments including chemotherapy, hormone therapy, immunotherapy or targeted therapies »

We have added the details in table 1.

“Results: toxicity”

  • Lines 223-224 “The actuarial risk of radionecrosis after repeated SRT was 25 % at 1 and 2 years, respectively”: The percentage is the same at 1year and at 2 years? It is not clear.

Yes, the percentage is the same for both. We have modified the sentence to make it clearer.

  • Line 229 “(…) systemic treatment concomitant (p = 0.04) were predictive of radionecrosis”: authors could clarify if there is a correlation between a specific type of systemic treatment (immunotherapy, targeted therapy, chemotherapy) and the incidence of radiation necrosis and/or symptomatic radiation necrosis.

We have detailed in the tables 4-5 the results by type of systemic treatment. No result is significant, this could be explained by the small number of patients

  • Is there any correlation between radiation necrosis and prior WBRT?

We have added this result in the table. We did not find any correlation

Reviewer 2 Report

This is a well conducted analysis of the value of stereotactic reirradiation in patients with miscellaneous brain metastases, mostly from NSCLC and breast cancer.

The minimum limitation of this paper is the lack of information on druggable molecular subgroups within NSCLC, breast cancer and melanoma. The authors should :

1. develop a Table reporting the number of patients in the major subgroups : for instance EGFR-mutant tumors, ALK rearranged tumors, etc..

2.  analyze whether the aforementioned patients received concurrently to reirradiation any new targeted agents, and in these cases analyze separately the efficacy in comparison to patients who had reirradiation alone.

3. in the Introduction, at line 49, add to reference 12 the reference of Soffietti et al, 2020 (Soffietti R, Ahluwalia M, Lin N, Rudà R. Management of brain metastases according to molecular subtypes. Nat Rev Neurol. 2020 Oct;16(10):557-574. doi: 10.1038/s41582-020-0391-x. Epub 2020 Sep 1. PMID: 32873927.).

Author Response

This is a well conducted analysis of the value of stereotactic reirradiation in patients with miscellaneous brain metastases, mostly from NSCLC and breast cancer.

The minimum limitation of this paper is the lack of information on druggable molecular subgroups within NSCLC, breast cancer and melanoma. The authors should :

  1. develop a Table reporting the number of patients in the major subgroups : for instance EGFR-mutant tumors, ALK rearranged tumors, etc..

We have added the details in table 1

  1. analyze whether the aforementioned patients received concurrently to reirradiation any new targeted agents, and in these cases analyze separately the efficacy in comparison to patients who had reirradiation alone.

We have detailed in the tables 4-5 the results by type of systemic treatment. No result is significant, this could be explained by the small number of patients

  1. in the Introduction, at line 49, add to reference 12 the reference of Soffietti et al, 2020 (Soffietti R, Ahluwalia M, Lin N, Rudà R. Management of brain metastases according to molecular subtypes. Nat Rev Neurol. 2020 Oct;16(10):557-574. doi: 10.1038/s41582-020-0391-x. Epub 2020 Sep 1. PMID: 32873927.).

We have added the reference

Round 2

Reviewer 2 Report

The authors have acceptablky revised the manuscript